# Prevalence and predictors of antibiotic prescription among patients hospitalized with viral lower respiratory tract infections in Southern Province, Sri Lanka

**Perla G. Medrano**[1,2☯*], **Nayani Weerasinghe**[3☯], **Ajith Nagahawatte**[2,3], **Sky Vanderburg**[4], **Lawrence P. Park**[1,2], **Gaya B. Wijayaratne**[3], **Vasantha Devasiri**[3], **Buddhika Dilshan**[3], **Tianchen Sheng**[1,2], **Ruvini Kurukulasooriya**[3], **Jack Anderson**[1], **Bradly P. Nicholson**[5], **Christopher W. Woods**[1,2], **Champica K. Bodinayake**[2,3‡], **L. Gayani Tillekeratne**[1,2,3‡]

1 Duke University, Durham, North Carolina, United States of America, 2 Duke Global Health Institute, Durham, North Carolina, United States of America, 3 Faculty of Medicine, University of Ruhuna, Galle, Sri Lanka, 4 University of California, San Francisco, California, United States of America, 5 Institute for Medical Research, Durham, North Carolina, United States of America

☯ These authors contributed equally to this work.
‡ CKB and LGT also contributed equally to this work.
* perla.medrano@duke.edu

## Abstract

### Background

Antimicrobial overprescription is common for lower respiratory tract infections (LRTI), as viral and bacterial infections generally present with similar clinical features. Overprescription is associated with downstream antimicrobial resistance. This study aims to identify the prevalence and predictors of antibiotic prescription among patients hospitalized with viral LRTI.

### Methods

A prospective cohort study was conducted among patients aged ≥1 year hospitalized with viral LRTI in a tertiary care hospital in Southern Province, Sri Lanka from 2018–2021. Demographic, clinical, and laboratory data were recorded. Nasopharyngeal and blood samples were collected for multiplex polymerase chain reaction testing for 21 respiratory pathogens and procalcitonin (PCT) detection, respectively. Demographic and clinical features associated with antibiotic prescription were identified using Chi Square and t-tests; significant variables (p<0.05) were further included in multivariable logistic regression models. The potential impact of biomarker testing on antibiotic prescription was simulated using standard c-reactive protein (CRP) and PCT cut-offs.

### Results

Of 1217 patients enrolled, 438 (36.0%) had ≥1 respiratory virus detected, with 48.4% of these patients being male and 30.8% children. Influenza A (39.3%) and human rhinovirus/ enterovirus (28.3%) were most commonly detected. A total of 114 (84.4%) children and 266

**Data Availability Statement:** Data cannot be shared publicly because public deposition would breach compliance with the protocol approved by the Ethics Review Committee (ERC), Faculty of Medicine, University of Ruhuna. Data are available from the ERC (contact via telephone: 0912234801/ 803 extension 161 or e-mail: ethics@med.ruh.ac. lk) for researchers who meet the criteria for access to confidential data.

**Funding:** This study was funded by a grant from the National Institutes of Health Fogarty Training Grant, (https://www.fic.nih.gov/Funding) #D4 TW009337 (LGT); Thrasher Research Foundation Early Career Award (https://www.thrasherresearch. org/early-career-award?lang=eng) (SV); National Institutes of Allergy and Infectious Diseases (https://www.niaid.nih.gov/) #K2AI125677 (LGT); and Duke Hubert-Yeargan Center for Global Health (https://hyc.globalhealth.duke.edu). The funders did not play any role in the study design, data collection and analysis, decision to publish, or preparation of the manuscript.

**Competing interests:** The authors have declared that no competing interests exist.

(87.8%) adults with respiratory viruses were treated with antibiotics. Among children, neutrophil percentage (median 63.6% vs 47.6%, p = 0.04) was positively associated with antibiotic prescription. Among adults, headache (60.6% vs 35.1%, p = 0.003), crepitations/ crackles (55.3% vs 21.6%, p<0.001), rhonchi/wheezing (42.9% vs 18.9%, p = 0.005), and chest x-ray opacities (27.4% vs 8.1%, p = 0.01) were associated with antibiotic prescription. Access to CRP and procalcitonin test results could have potentially decreased inappropriate antibiotic prescription in this study by 89.5% and 83.3%, respectively.

## Conclusions

High proportions of viral detection and antibiotic prescription were observed among a large inpatient cohort with LRTI. Increased access to point-of-care biomarker testing may improve antimicrobial prescription.

## Introduction

Lower respiratory tract infection (LRTI) remains the world's deadliest communicable disease and a top five leading cause of infectious admissions and disability-adjusted life years, leading to a total of 489 million cases and 2.5 million deaths in 2019 [1, 2]. LRTI is comprised of conditions including pneumonia, bronchitis, and infective exacerbations of chronic obstructive pulmonary disease (COPD) and asthma [3]. LRTI is caused primarily by bacterial and viral pathogens, and less commonly by fungal organisms; however, viral LRTI is often more commonly detected than bacterial LRTI among both children and adults [4–7]. Both bacterial and viral LRTI generally present with similar clinical features, such as fever, cough, chest pain, and sputum production [3]. The similarity of these clinical manifestations has posed a great challenge for physicians worldwide in managing and distinguishing bacterial and viral LRTI. For this reason, physicians often overprescribe antibiotics for viral LRTI [8–10]. Alternatively, failure to effectively treat LRTI conditions can result in a higher risk for severe complications, prolonged course of symptoms, secondary infections, and mortality, especially among vulnerable populations such as the immunosuppressed, elderly, or those with pre-existing health conditions [3, 4].

Antibiotic overprescription and misprescription contribute to an emerging top ten global health threat: antimicrobial resistance (AMR) [11, 12]. AMR occurs when bacteria, viruses, and other pathogens adapt to antimicrobial drugs and evolve into strong, drug-resistant organisms that decrease antimicrobial drug effectiveness and impede the treatment of infectious diseases [11]. While AMR caused a global burden of approximately 4.95 million deaths in 2019, it has been shown to have a disproportionate impact in low- and middle-income countries (LMICs), causing some of the highest rates of death in South Asia [8].

LRTI may be diagnosed through a combination of clinical assessment, including the evaluation of clinical features, medical history, and physical exam findings, as well as through chest radiography. Specific diagnostic tests to differentiate between bacterial and viral etiologies include pathogen-based diagnostics, such as sputum Gram stain and culture, blood cultures, serologic studies, antigen detection tests, and nucleic acid amplification tests [13, 14]. However, the sensitivity and specificity of these pathogen-based diagnostics may be limited. Invasive procedures–such as bronchoscopic bronchoalveolar lavage, transthoracic needle aspiration, or protected specimen brush–may provide a more accurate etiology but are

infrequently performed due to their invasive nature [10]. Biomarkers, such as C-reactive protein (CRP) and procalcitonin (PCT), may also be used as an adjunct to identify the class of infection (bacterial versus viral) and to improve antimicrobial use [10].

In Sri Lanka, a LMIC in South Asia, LRTI is a common reason for hospitalization among both children and adults. Distinguishing viral and bacterial LRTI can be difficult due to insufficient resources and diagnostics [9, 15, 16]. In 2018, a Sri Lankan study reported that 30.1% of hospitalized patients were potentially inappropriately prescribed antibiotics, for which LRTI was the most common indication for prescription [9]. Further, a 2021 study found high proportions of antibiotic prescription among outpatients in Sri Lanka, of whom only 4.1% of patients presenting with respiratory symptoms underwent diagnostic testing [17]. Despite concerns related to the high prevalence of LRTI and inappropriate antibiotic prescription in Sri Lanka, there is limited information on predictors of antibiotic prescription among patients with viral LRTI.

To improve antibiotic prescribing practices and appropriately inform future antibiotic stewardship interventions, it is critical to understand antibiotic prescription patterns and features associated with antibiotic prescription among patients with viral LRTI. The aim of this study was to identify the prevalence and predictors of antibiotic prescription among children and adults hospitalized with viral LRTI in Sri Lanka.

## Materials and methods

### Setting

The Respiratory Infection Severity and Etiology in Sri Lanka (RISE-SL) study was conducted in a large, 1,800-bed, public tertiary care hospital in Southern Province, Sri Lanka from April 2018 to October 2021. The hospital is administered by the Ministry of Health and offers free outpatient and inpatient services, including medical, pediatric, and surgical care, to a catchment population of 3.5 million.

### Participants

Consecutive patients ≥1 year old admitted with acute LRTI were identified for enrollment within 48 hours of admission. Patients were eligible if they met an age-specific case definition for LRTI and had an acute illness for a duration of <14 days. For patients ≥5 years, the case definition included 1) evidence of acute respiratory illness, as indicated by at least one sign or symptom (new cough, sputum production, chest pain, dyspnea, tachypnea [>25 breaths/minute], abnormal lung examination, or need for mechanical ventilation) and 2) evidence of acute infection, as indicated by at least one sign or symptom (reported fever or chills, documented fever or hypothermia, leukocytosis, leukopenia, or new altered mental status). Chest x-ray imaging within 48 hours of admission was required for eligibility for patients ≥5 years. For patients <5 years of age, the case definition for LRTI included 1) evidence of acute respiratory illness, as indicated by cough or difficulty breathing and 2) evidence of acute infection, as indicated by at least one of the following: lower chest wall indrawing, tachypnea (≥40 breaths per minute), oxygen saturation <90%, central cyanosis, severe respiratory distress, inability to drink or breastfeed, vomiting everything, altered consciousness, or convulsions. Chest x-rays were not required in patients <5 years of age. Patients were not eligible to participate in this study if they were outpatients, hospitalized within the past 28 days, unable or unwilling to provide consent or biological samples, had known or suspected infections at other anatomic sites requiring antibacterial therapy, or had any specific condition that could have precluded participation and affected subject safety according to study physicians or clinical providers. The study recruitment period spanned from April 24, 2018, to January 19, 2021.

## Data collection

A standardized questionnaire on demographic and clinical information was administered to all patients by trained research assistants. A nasopharyngeal sample and blood sample were collected at enrollment and stored at -80˚C until used for testing. Patients were followed longitudinally during hospitalization and information relating to laboratory and radiographic results, treatments administered, and clinical outcomes were obtained from the medical record. Clinical decision-making was conducted by the routine care providers and was not influenced by research personnel.

## Laboratory testing

Nasopharyngeal samples were tested using the Luminex NxTAG Respiratory Pathogen Panel (Luminex Corporation, Austin, TX, USA) [18]. This panel detects 3 bacterial pathogens–*Chlamydophila pneumoniae*, *Mycoplasma pneumoniae*, and *Legionella pneumophila*–and 18 viral pathogens, including adenovirus, bocavirus, coronavirus types 229E, HKU1, NL63, and OC43, influenza types A, A H1, A H3, and B, human metapneumovirus, parainfluenza types 1–4, respiratory syncytial virus types A and B, and human rhinovirus/enterovirus; human rhinovirus and enterovirus are closely related members of the Picornaviridae family, therefore, the assay cannot reliably differentiate them due to their genetic similarity [19]. This panel was chosen for its diagnostic capabilities across a wide array of targets, capacity to process up to 96 samples per run, enhanced sensitivity for certain viruses and detection of coinfections, and benefits such as reduced manual labor and lower consumption of consumable and reagents when compared with alternative methods [20–22]. Testing for SARS-Coronavirus-2 was conducted using the Centers for Disease Control and Prevention (CDC) SARS-CoV-2 assay on an AB7500 Fast DX (Applied Biosystems). PCT testing was performed on serum samples using the VIDAS BRAHMS PCT kit using the Enzyme-Linked Fluorescent Assay technique via the Mini-VIDAS platform (bioMérieux, Marcy-l'Étoile, France).

## Statistical analysis

Data analyses were conducted using R Statistical Software (R Core Team 2022). Descriptive statistical analysis was conducted to determine the prevalence of respiratory viral positivity and antibiotic prescription. Bivariable analyses were conducted using Chi Square and t-tests to identify the association between sociodemographic characteristics, clinical features, and physical, laboratory, and radiographic examination results and antibiotic prescriptions among patients with viral detection. Variables which were statistically significant ($p<0.05$) on bivariable analysis were checked for collinearity and included in multivariable logistic regression models. Two separate multivariable models were created for children and adults. Backward elimination methods were implemented, whereby full models were initially fitted with all predictor variables, and variables with the greatest p-values were gradually removed at each step until all variables met the significance threshold ($p<0.05$). Excluded variables were added back to the models one at a time to verify that they were not significant.

Chest x-ray result (normal/ abnormal) was excluded as a variable from the multivariable logistic regression model for adults, since presence of opacity/ consolidation on x-ray was included as a variable and was thought to be more specific for bacterial infection. Abnormal x-rays could include findings such as interstitial pattern, which is seen with viruses. To prevent model instability, diarrhea and x-ray opacity variables were also excluded from the multivariable model for children, as both had frequency percentages of less than 5% among those with no antibiotic prescription.

The potential impact of PCT and CRP testing on antibiotic prescription, assuming clinical guidelines were followed and PCT test results were available to the physicians at the time of clinical decision-making, were simulated individually using standard PCT and CRP cut-offs of >0.25 ng/mL and $\geq$ 20 mg/L, respectively, to indicate the likelihood of clinically relevant bacterial infection.

## Ethical procedures

Ethical approval for this research study was acquired from the Duke University Institutional Review Board (USA) and Ethical Review Committee of the Faculty of Medicine, University of Ruhuna (Sri Lanka). Written informed consent was obtained from patients $\geq$18 years of age and the parents or guardians of patients 1–17 years of age. Written assent was obtained from pediatric patients aged 12–17 years.

## Results

A total of 1217 patients were enrolled during the study period, including 190 (15.6%) children under 18 years of age, 614 (50.6%) male patients, and 600 (49.4%) female patients. Of the 1217 patients, 438 (36.0%) had at least one respiratory virus detected. The most commonly detected viruses were influenza A (39.3%), human rhinovirus/ enterovirus (HRV/HEV; 28.3%), and respiratory syncytial virus A (RSV A; 10.3%) (Table 1). Among children, the most commonly detected viruses were influenza A (7.5%), HRV/HEV (7.3%), and bocavirus (5.9%); and those among adults were influenza A (31.7%), HRV/HEV (21.0%), and human metapneumovirus (5.3%). Bocavirus (5.9% vs 3.7%, p<0.001) and RSV A (5.3% vs 5.0%, p = 0.002) were detected more commonly in children than adults. Influenza A was detected less commonly in children

**Table 1. Multiplex viral panel results of overall cohort of 1217 patients with lower respiratory tract infection and viral detection in Sri Lanka, 2018–2021 (n = 438).**

|  | Overall n (%) | Children n (%) | Adults n (%) | P value [a] |
|---|---|---|---|---|
| Adenovirus | 34 (7.8) | 17 (3.9) | 17 (3.9) | **0.01** |
| Bocavirus | 42 (9.6) | 26 (5.9) | 16 (3.7) | **<0.001** |
| Coronavirus 229E | 6 (1.4) | 2 (0.5) | 4 (0.9) | 0.89 |
| Coronavirus HKU1 | 1 (0.2) | 1 (0.2) | 0 (0.0) | 0.13 |
| Coronavirus NL63 | 5 (1.1) | 3 (0.7) | 2 (0.5) | 0.16 |
| Coronavirus OC43 | 10 (2.3) | 1 (0.2) | 9 (2.1) | 0.15 |
| Influenza A | 172 (39.3) | 33 (7.5) | 139 (31.7) | **<0.001** |
| Influenza B | 29 (6.6) | 11 (2.5) | 18 (4.1) | 0.39 |
| Human metapneumovirus | 42 (9.6) | 19 (4.3) | 23 (5.3) | 0.03 |
| Parainfluenza 1 | 5 (1.1) | 1 (0.2) | 4 (0.9) | 0.60 |
| Parainfluenza 2 | 1 (0.2) | 0 (0.0) | 1 (0.2) | 0.50 |
| Parainfluenza 3 | 16 (3.7) | 5 (1.1) | 11 (2.5) | 0.97 |
| Parainfluenza 4 | 3 (0.7) | 1 (0.2) | 2 (0.5) | 0.92 |
| Respiratory Syncytial Virus A | 45 (10.3) | 23 (5.3) | 22 (5.0) | **0.002** |
| Respiratory Syncytial Virus B | 13 (3.0) | 5 (1.1) | 8 (1.8) | 0.54 |
| Human rhinovirus/ enterovirus | 124 (28.3) | 32 (7.3) | 92 (21.0) | 0.15 |
| Total multiplex viral panel results | 548 | 180 | 368 |  |

Some patients had more than one virus detected.

[a] P values in bold are significant at <0.05.

than adults (7.5% vs 31.7%, p<0.001). None of the patients tested positive for COVID-19 in our study.

The results presented in the remainder of the manuscript focus on the 438 patients with viral detection.

## Antibiotic prescription among patients with viral detection

Of the 438 patients with viral detection, 135 (30.8%) were children and 212 (48.4%) were male; the median age was 4 years (IQR 2–7) for children and 61 years (IQR 48–70) for adults (Table 2). Overall, 380 (86.8%) patients with respiratory viruses were treated with antibiotics during hospitalization, including 114 (84.4%) children and 266 (87.8%) adults (Table 3). There was no difference in proportion receiving antibiotic prescription between children and adults (p = 0.34).

Among the 380 (86.8%) patients with viral detection who received antibiotics, the most commonly prescribed antibiotics were amoxicillin and clavulanic acid (44.5%), third-

**Table 2. Bivariable analysis of demographic and chronic medical condition features associated with antibiotic prescription among patients with LRTI and viral detection (n = 438) in Sri Lanka, 2018–2021.**

|  | Antibiotic prescription n (%) | No antibiotic prescription n (%) | P value [a] |
|---|---|---|---|
| **Sex** |  |  |  |
| Male | 179 (47.2) | 33 (57.9) | 0.13 |
| **Age (years)** |  |  |  |
| <18 | 114 (30.0) | 21 (36.2) | 0.05 |
| 18–64 | 148 (39.0) | 28 (48.3) |  |
| ≥65 | 118 (31.1) | 9 (15.5) |  |
| **Exposure to tobacco smoke** |  |  |  |
| Never | 283 (78.4) | 35 (67.3) | 0.16 |
| Current | 13 (3.6) | 5 (9.6) |  |
| Prior | 63 (17.5) | 12 (23.1) |  |
| Other | 2 (0.6) | 0 (0.0) |  |
| **Chronic medical conditions** |  |  |  |
| Asthma | 127 (33.4) | 14 (24.1) | 0.16 |
| Bronchiectasis | 15 (4.0) | 0 (0.0) | 0.12 |
| Chronic obstructive pulmonary disease | 18 (4.7) | 2 (3.5) | 0.66 |
| Hypertension | 72 (19.0) | 9 (15.5) | 0.53 |
| Ischemic heart disease | 37 (9.7) | 4 (6.9) | 0.49 |
| Chronic heart failure | 3 (0.8) | 1 (1.7) | 0.49 |
| Diabetes | 38 (10.0) | 11 (19.0) | **0.04** |
| Chronic kidney disease | 12 (3.2) | 4 (6.9) | 0.16 |
| Chronic liver disease | 4 (1.1) | 1 (1.7) | 0.65 |
| Anemia / thalassemia | 3 (0.8) | 1 (1.7) | 0.49 |
| Cancer [b] | 2 (0.5) | 0 (0.0) | 0.58 |
| Human immunodeficiency virus | 0 (0.0) | 0 (0.0) | - |
| Pulmonary tuberculosis | 6 (1.6) | 0 (0.0) | 0.34 |
| Immunosuppression [c] | 12 (3.2) | 1 (1.7) | 0.55 |

[a] P values in bold are significant at <0.05.

[b] Active for the past 5 years.

[c] Immunosuppression in the past 30 days.

**Table 3. Antibiotic prescription during hospitalization among patients with lower respiratory tract infection and viral detection (n = 438) in Sri Lanka, 2018–2021.**

| | Overall n (%) | Children n (%) | Adults n (%) | P value [a] |
|---|---|---|---|---|
| **Antibiotic prescription** | | | | |
| During hospitalization | 380 (86.8) | 114 (26.0) | 266 (60.7) | 0.34 |
| **Antibiotic types** | | | | |
| Amoxicillin/ ampicillin | 7 (1.6) | 3 (0.7) | 4 (0.9) | 0.49 |
| Dicloxacillin/ nafcillin/ oxacillin | 2 (0.5) | 0 (0.0) | 2 (0.5) | 0.34 |
| 1st generation cephalosporin (*i.e.*, cephalexin) | 2 (0.5) | 2 (0.5) | 0 (0.0) | **0.03** |
| 2nd generation cephalosporin (*i.e.*, cefuroxime) | 33 (7.5) | 18 (4.1) | 15 (3.4) | **0.002** |
| 3rd generation cephalosporin (*i.e.*, ceftriaxone) | 135 (30.8) | 53 (12.1) | 82 (18.7) | **0.01** |
| Amoxicillin & clavulanic acid | 195 (44.5) | 29 (6.6) | 166 (37.9) | **<0.001** |
| Piperacillin/ tazobactam | 8 (1.8) | 2 (0.5) | 6 (1.4) | 0.72 |
| Carbapenem (*i.e.*, meropenem) | 18 (4.1) | 6 (1.4) | 12 (2.8) | 0.81 |
| Aztreonam | 0 (0.0) | 0 (0.0) | 0 (0.0) | - |
| Vancomycin | 11 (2.5) | 8 (1.8) | 3 (0.7) | **0.002** |
| Metronidazole | 8 (1.8) | 1 (0.2) | 7 (1.6) | 0.26 |
| Trimethoprim & sulfamethoxazole | 0 (0.0) | 0 (0.0) | 0 (0.0) | - |
| Clindamycin | 6 (1.4) | 0 (0.0) | 6 (1.4) | 0.10 |
| Erythromycin/ azithromycin | 42 (9.6) | 24 (5.5) | 18 (4.1) | **<0.001** |
| Clarithromycin | 133 (30.4) | 33 (7.5) | 100 (22.8) | 0.07 |
| Tetracycline/ doxycycline | 6 (1.4) | 0 (0.0) | 6 (1.4) | 0.10 |
| Fluoroquinolone (*i.e.*, ciprofloxacin) | 23 (5.3) | 13 (3.0) | 10 (2.3) | **0.006** |
| Aminoglycoside (*i.e.*, gentamicin) | 1 (0.2) | 1 (0.2) | 0 (0.0) | 0.13 |
| Total antibiotic prescriptions | 630 | 193 | 437 | |

[a] P values in bold are significant at <0.05.

generation cephalosporins (30.8%), and clarithromycin (30.4%) (Table 3). These three antibiotics were also the most commonly prescribed among pediatric and adult patients, when analyzed individually. First-generation cephalosporins (0.5% vs 0.0%, p = 0.03), second-generation cephalosporins (4.1% vs 3.4%, p = 0.002), vancomycin (1.8% vs 0.7%, p = 0.002), erythromycin/ azithromycin (5.5% vs 4.1%, p<0.001), and fluoroquinolones (3.0% vs 2.3%, p = 0.006) were prescribed more commonly in children than adults, respectively. Amoxicillin and clavulanic acid (6.6% vs 37.9%, p<0.001) and third-generation cephalosporins (12.1% vs 18.7%, p = 0.01) were prescribed less commonly among children than adults.

### Demographic and clinical features associated with antibiotic prescription

On bivariable analysis, there was a non-significant trend related to the likelihood of antibiotic prescription based on age (p = 0.05), with patients aged <65 years being less likely to be prescribed antibiotics than other patients (68.9% vs 84.5%, Table 2). Patients with diabetes were less likely to be prescribed antibiotics than other patients (10.0% vs 19.0%, p = 0.04).

The most common clinical symptom upon admission was cough (411, 10.0%), followed by fever (380, 9.0%) and chills (374, 9.0%) (Table 4). Pediatric patients presenting with diarrhea were significantly more likely to be prescribed antibiotics than other patients (22.4% vs 0.0%, p = 0.02); however, those presenting with wheezing were significantly less likely to be prescribed antibiotics than other patients (19.6% vs 45.0%, p = 0.01). Adult patients presenting with cough (p = 0.03) or headache (p = 0.003) were significantly more likely to be prescribed antibiotics than other patients. Fever was not significantly associated with antibiotic

**Table 4. Bivariable analysis of clinical symptoms, physical exam, and clinical diagnosis features associated with antibiotic prescription among patients with LRTI and viral detection (n = 438) in Sri Lanka, 2018–2021.**

| | Children (n = 135) | | | Adults (n = 303) | | |
|---|---|---|---|---|---|---|
| | Antibiotic prescription n (%) | No antibiotic prescription n (%) | P value [a] | Antibiotic prescription n (%) | No antibiotic prescription n (%) | P value [a] |
| **Clinical symptoms** | | | | | | |
| Fever | 109 (96.5) | 19 (90.5) | 0.22 | 225 (84.9) | 27 (73.0) | 0.07 |
| Cough | 107 (93.9) | 19 (90.5) | 0.57 | 253 (95.5) | 32 (86.5) | **0.03** |
| Ear pain | 10 (9.2) | 0 (0.0) | 0.15 | 32 (12.3) | 1 (2.8) | 0.09 |
| Headache | 47 (43.5) | 8 (40.0) | 0.77 | 160 (60.6) | 13 (35.1) | **0.003** |
| Chills | 47 (43.1) | 10 (47.6) | 0.70 | 133 (50.4) | 16 (43.2) | 0.42 |
| Sneezing | 55 (49.6) | 8 (38.1) | 0.34 | 71 (27.5) | 14 (37.8) | 0.20 |
| Runny nose | 62 (56.4) | 13 (61.9) | 0.64 | 97 (37.5) | 9 (25.0) | 0.14 |
| Nasal Congestion | 41 (37.3) | 11 (52.4) | 0.19 | 68 (26.2) | 8 (21.6) | 0.55 |
| Sore throat | 24 (21.6) | 6 (28.6) | 0.49 | 76 (29.1) | 11 (30.6) | 0.86 |
| Scratchy/itchy throat | 5 (4.5) | 2 (9.5) | 0.35 | 41 (15.9) | 3 (8.3) | 0.23 |
| Hoarse/voice change | 24 (21.8) | 6 (28.6) | 0.50 | 100 (38.5) | 11 (29.7) | 0.30 |
| Short of breath | 63 (56.3) | 11 (52.4) | 0.74 | 168 (63.9) | 22 (61.1) | 0.75 |
| Pain with inspiration | 21 (19.7) | 2 (10.0) | 0.32 | 81 (31.0) | 12 (33.3) | 0.78 |
| Chest pain | 12 (11.1) | 2 (9.5) | 0.83 | 69 (26.1) | 9 (25.0) | 0.88 |
| Decreased hearing | 2 (1.9) | 0 (0.0) | 0.53 | 13 (5.0) | 3 (8.1) | 0.43 |
| Pain behind eyes | 7 (6.5) | 0 (0.0) | 0.23 | 40 (15.6) | 5 (13.5) | 0.74 |
| Itchy/watery eyes | 31 (27.9) | 6 (28.6) | 0.95 | 51 (19.6) | 6 (16.2) | 0.62 |
| Stiff neck | 2 (1.8) | 0 (0.0) | 0.54 | 11 (4.3) | 3 (8.1) | 0.31 |
| Malaise/fatigue | 41 (38.7) | 9 (42.9) | 0.72 | 151 (58.3) | 15 (42.9) | 0.08 |
| Decreased appetite | 74 (68.5) | 12 (57.1) | 0.31 | 178 (68.2) | 25 (67.6) | 0.94 |
| Joint pain | 24 (21.6) | 2 (9.5) | 0.20 | 72 (27.4) | 10 (27.0) | 0.96 |
| Muscle pain | 19 (17.6) | 1 (4.8) | 0.14 | 55 (21.0) | 10 (27.0) | 0.40 |
| Rash/flushing | 3 (2.8) | 0 (0.0) | 0.44 | 2 (0.8) | 1 (2.8) | 0.26 |
| Abdominal pain | 26 (23.9) | 7 (33.3) | 0.36 | 37 (14.2) | 3 (8.1) | 0.31 |
| Vomiting | 70 (62.5) | 9 (42.9) | 0.09 | 69 (26.2) | 7 (18.9) | 0.34 |
| Diarrhea | 24 (22.4) | 0 (0.0) | **0.02** | 34 (13.1) | 5 (13.9) | 0.89 |
| Painful urination | 5 (4.5) | 1 (4.8) | 0.96 | 23 (8.8) | 3 (8.1) | 0.89 |
| Decreased urination | 9 (8.2) | 1 (5.0) | 0.62 | 21 (8.1) | 2 (5.7) | 0.63 |
| Bleeding | 0 (0.0) | 0 (0.0) | - | 4 (1.5) | 1 (2.8) | 0.58 |
| Wheezing | 20 (19.6) | 9 (45.0) | **0.01** | 96 (40.0) | 12 (36.4) | 0.69 |
| Pregnancy | 0 (0.0) | 0 (0.0) | - | 2 (0.9) | 1 (3.3) | 0.25 |
| **Physical exam findings within 24 hours of admission** | | | | | | |
| Crepitations/crackles | 34 (29.8) | 10 (47.6) | 0.11 | 147 (55.3) | 8 (21.6) | **<0.001** |
| Rhonchi/wheezing | 41 (36.0) | 8 (38.1) | 0.85 | 114 (42.9) | 7 (18.9) | **0.005** |
| Conjunctival injection/ suffusion | 1 (0.9) | 0 (0.0) | 0.67 | 5 (1.9) | 1 (2.8) | 0.73 |
| Jaundice | 1 (0.9) | 1 (5.0) | 0.17 | 2 (0.8) | 0 (0.0) | 0.60 |
| Throat erythema | 3 (2.7) | 1 (5.0) | 0.59 | 2 (0.8) | 0 (0.0) | 0.60 |
| Nodes | 4 (4.6) | 2 (10.5) | 0.38 | 2 (0.9) | 0 (0.0) | 0.60 |
| Dullness | 0 (0.0) | 0 (0.0) | - | 4 (1.5) | 0 (0.0) | 0.45 |
| Absent breath sounds | 1 (0.9) | 0 (0.0) | 0.67 | 0 (0.0) | 0 (0.0) | - |
| Nasal flaring/grunting | 0 (0.0) | 0 (0.0) | - | 3 (1.1) | 0 (0.0) | 0.52 |

*(Continued)*

**Table 4.** (Continued)

| | Children (n = 135) | | | Adults (n = 303) | | |
|---|---|---|---|---|---|---|
| | Antibiotic prescription n (%) | No antibiotic prescription n (%) | P value [a] | Antibiotic prescription n (%) | No antibiotic prescription n (%) | P value [a] |
| Central cyanosis | 1 (0.9) | 0 (0.0) | 0.67 | 0 (0.0) | 0 (0.0) | - |
| Chest wall indrawing | 5 (4.4) | 2 (9.5) | 0.33 | 3 (1.1) | 0 (0.0) | 0.52 |
| Enlarged liver | 2 (1.8) | 0 (0.0) | 0.54 | 2 (0.8) | 1 (2.9) | 0.24 |
| Tender abdomen | 3 (2.7) | 0 (0.0) | 0.45 | 7 (2.7) | 1 (2.9) | 0.94 |
| Enlarged spleen | 0 (0.0) | 0 (0.0) | - | 1 (0.4) | 0 (0.0) | 0.71 |
| Free fluid | 0 (0.0) | 1 (5.0) | **0.02** | 3 (1.2) | 0 (0.0) | 0.53 |
| Drowsy/confused/coma | 104 (92.9) | 2 (10.0) | 0.81 | 6 (2.3) | 1 (2.8) | 0.40 |
| Rash | 0 (0.0) | 0 (0.0) | - | 5 (1.9) | 0 (0.0) | 0.41 |
| **Clinical diagnosis on admission** | | | | | | |
| LRTI | 84 (73.7) | 9 (42.9) | **0.005** | 180 (67.7) | 16 (43.2) | **0.004** |
| URTI | 8 (7.0) | 2 (9.5) | 0.69 | 8 (3.0) | 2 (5.4) | 0.44 |
| Pulmonary tuberculosis | 0 (0.0) | 0 (0.0) | - | 3 (1.1) | 0 (0.0) | 0.52 |
| COPD exacerbation | 1 (0.9) | 0 (0.0) | 0.67 | 22 (8.3) | 2 (5.4) | 0.55 |
| Asthma exacerbation | 9 (7.9) | 4 (19.1) | 0.11 | 37 (13.9) | 5 (13.5) | 0.95 |
| Dengue | 2 (1.8) | 0 (0.0) | 0.54 | 4 (1.5) | 1 (2.7) | 0.59 |
| Leptospirosis | 0 (0.0) | 0 (0.0) | - | 2 (0.8) | 0 (0.0) | 0.60 |
| Typhus | 0 (0.0) | 0 (0.0) | - | 0 (0.0) | 0 (0.0) | - |

Bivariable analysis of features relating to clinical symptoms, physical examination within 24 hours of admission, and clinical diagnosis upon admission associated with antibiotic prescription during hospitalization among patients with lower respiratory tract infection and viral detection (n = 438) in Sri Lanka, 2018–2021.

LRTI = lower respiratory tract infection; URTI = upper respiratory tract infection; COPD = chronic obstructive pulmonary disease.

[a] P values in bold are significant at <0.05.

prescription among either children or adults; however, there was a non-significant trend related to the likelihood of antibiotic prescription among adult patients presenting with fever (p = 0.07).

The most common physical examination finding was crepitations/ crackles (199, 43.6%), followed by rhonchi/ wheezing (170, 37.3%) and drowsiness/ confusion (17, 3.7%) (Table 4). Pediatric patients presenting with free fluid were significantly more likely to be prescribed antibiotics than other patients (p = 0.02); however only 1 (0.8%) patient was found to present with free fluid. Adult patients presenting with crepitations/crackles (55.3% vs 21.6%, p<0.001) or rhonchi/ wheezing (42.9% vs 18.9%, p = 0.005) were significantly more likely to be prescribed antibiotics than other patients.

The most common clinical diagnosis upon admission was LRTI (289, 66.0%), followed by asthma exacerbation (55, 12.6%) and COPD exacerbation (25, 5.7%) (Table 4). Both pediatric (p = 0.005) and adult (p = 0.004) patients with an admission clinical diagnosis of LRTI were significantly more likely to be prescribed antibiotics than other patients.

## Routine laboratory testing results associated with antibiotic prescription

We looked at laboratory results within the first 48 hours of hospitalization as this is a critical period for antibiotic prescription decision-making. Neutrophil percentage (p = 0.04) and ESR (p<0.001) were significantly associated with antibiotic prescription (Table 5); patients who were prescribed with antibiotics had median values of 71.8 (IQR 60.1–81.4) for neutrophil

**Table 5. Bivariable analysis of laboratory test and chest radiograph features associated with antibiotic prescription among patients with LRTI and viral detection (n = 438) in Sri Lanka, 2018–2021.**

| | Children (n = 135) | | | Adults (n = 303) | | |
|---|---|---|---|---|---|---|
| | Antibiotic prescription | No antibiotic prescription | P value [a] | Antibiotic prescription | No antibiotic Prescription | P value [a] |
| | n (%) / median (IQR) | | | n (%) / median (IQR) | | |
| **Laboratory tests within 48 hours of admission** | | | | | | |
| Hemoglobin (g/dL) | 11.6 (10.9–12.6) | 11.8 (10.5–12.1) | 0.66 | 12.2 (11.2–13.5) | 13.2 (12.0–14.1) | 0.19 |
| Hematocrit (%) | 34 (32.4–36.9) | 34.5 (33.4–35.6) | 0.30 | 36.6 (34.2–40.1) | 39.4 (34.9–41.4) | 0.15 |
| Platelets ($10^{12}$/L) | 290 (239.0–370.0) | 255 (167.0–294.0) | 0.31 | 251.5 (189.5–310.3) | 225 (191.0–295.0) | 0.43 |
| WBC ($10^9$/L) | 11.6 (8.0–14.6) | 8.4 (6.5–9.9) | 0.07 | 10 (6.6–13.1) | 10.3 (7.2–11.3) | 0.72 |
| Neutrophils (%) | 63.6 (46.0–78.1) | 47.6 (31.0–62.6) | **0.04** | 74.3 (64.0–82.8) | 69.2 (63.4–82.6) | 0.38 |
| Leukocytosis ($10^9$/L) | 64 (60.4) | 3 (23.1) | **0.01** | 123 (50.2) | 17 (54.8) | 0.63 |
| AST / SGOT (U/L) | 37 (32.0–52.0) | 0 (0.0–0.0) | - | 28 (20.0–38.8) | 17.5 (15.5–31.5) | 0.97 |
| ALT / SGPT (U/L) | 14.5 (12.0–30.3) | 0 (0.0–0.0) | - | 24 (17.0–36.0) | 19.5 (12.8–32.3) | 0.53 |
| CRP (mg/L) [b] | 17 (7.0–37.5) | 30 (7.3–56.5) | 0.45 | 40 (17.0–74.0) | 18 (10.0–29.0) | **0.001** |
| ESR (mm/hr) [c] | 23.5 (18.8–50.0) | 0 (0.0–0.0) | - | 40 (22.0–66.0) | 14 (7.5–16.0) | **<0.001** |
| **Chest radiographs [d]** | | | | | | |
| Abnormal reading | 44 (55.7) | 4 (57.1) | 0.94 | 122 (46.9) | 9 (28.1) | **0.04** |
| Opacity / consolidation | 28 (24.6) | 1 (4.8) | **0.04** | 73 (27.4) | 3 (8.1) | **0.01** |
| Interstitial pattern | 6 (5.3) | 2 (9.5) | 0.45 | 15 (5.6) | 1 (2.7) | 0.45 |
| Nodules | 1 (0.9) | 1 (4.8) | 0.18 | 6 (2.3) | 0 (0.0) | 0.36 |
| Mass | 2 (1.8) | 1 (4.8) | 0.39 | 3 (1.1) | 0 (0.0) | 0.52 |
| Effusion | 58 (50.9) | 14 (66.7) | 0.18 | 32 (12.0) | 6 (16.2) | 0.47 |

IQR = interquartile range; WBC = white blood cell count; AST / SGOT = aspartate aminotransferase/ serum glutamic-oxaloacetic transaminase; ALT / SGPT = alanine aminotransferase/ serum glutamic-pyruvic transaminase; CRP = c-reactive protein; ESR = erythrocyte sedimentation rate; mm/hr = millimeters per hour.

[a] P values in bold are significant at <0.05.

[b] Data on CRP levels were only collected for 300 patients (82 children and 218 adults) during routine clinical care.

[c] Data on ESR levels were only collected for 114 patients (12 children and 102 adults) during routine clinical care.

[d] All 438 patients received chest radiographs (135 children and 303 adults).

percentage and 40 (IQR 21.0–64.0) for ESR. Among pediatric patients, laboratory test findings of neutrophil percentage (p = 0.04) were significantly associated with antibiotic prescription; children who were prescribed antibiotics had a median neutrophil percentage value of 63.6 (IQR 46.0–78.1) versus 47.6 (IQR 31.0–62.6) in those who were not. Pediatric patients presenting with leukocytosis (64, 60.4%) were significantly more likely to be prescribed with antibiotics than other patients (3, 23.1%, p = 0.01). Among adult patients, laboratory test results of CRP (p = 0.001) and ESR (p<0.001) were significantly associated with antibiotic prescription; adults who were prescribed antibiotics had CRP and ESR median values of 40mg/L (IQR 17.0–74.0) and 40mm/ hour (IQR 22.0–66.0), respectively.

## Chest radiograph features associated with antibiotic prescription

Overall, all 438 patients with viral detection (135 children and 303 adults) received chest radiographs. A total of 179 (47.4%) chest radiographs showed abnormal results, including those of 48 (35.6%) children and 131 (43.2%) adults (Table 5). Among patients with parenchymal abnormalities, 105 (24.0%) had opacities, 110 (25.1%) had effusions, and 24 (5.5%) had interstitial patterns. On bivariable analysis, pediatric patients with opacity/ consolidation on chest

x-ray were significantly more likely to be prescribed antibiotics than patients without this finding (24.6% vs 4.8%, p = 0.04). Adult patients presenting with abnormal readings (46.9% vs 28.1%, p = 0.04) or opacity/ consolidation (27.4% vs 8.1%, p = 0.01) were significantly more likely to be prescribed antibiotics than other patients.

## Multivariable logistic regression of features associated with antibiotic prescription

In the final multivariable model, neutrophil percentage was associated with higher odds of antibiotic prescription among pediatric patients (OR 1.04, 95% CI 1.01–1.07) (Table 6). Adult patients presenting with rhonchi/ wheezing on exam (OR 4.08, 95% CI 1.76–10.70), opacities on chest x-ray (OR 4.15, 95% CI 1.25–21.85), creptations/ crackles on exam (OR 4.94, 95% CI 2.15–12.74), or headache (OR 1.39, 95% CI 0.97–1.91) were more likely to be prescribed antibiotics.

## C-reactive protein (CRP) testing

Patients with viral detection had a median CRP level of 30.5 (IQR 11.2–66.3), with 17 (IQR 7.0–41.3) for children and 37 (IQR 15.0–72.8) for adults. On simulation using the standard CRP cut-off of $\geq 20$ mg/L to identify bacterial LRTI, having adhered to CRP test results may have reduced antibiotic prescription among physicians (Table 7). A level of 60% adherence to CRP-guided antibiotic prescription using CRP $\geq 20$ mg/L may have potentially reduced inappropriate antibiotic prescription by 33.6% in children and 16.7% in adults; 80% adherence may have potentially reduced inappropriate antibiotic prescription by 44.8% in children and 22.3% in adults; and 100% adherence may have potentially reduced inappropriate antibiotic prescription by 56.0% in children and 27.9% in adults. The potential reduction in antibiotic prescription was significantly greater among pediatric patients than adult patients at 60%, 80%, and 100% adherence levels.

## Procalcitonin (PCT) testing

Patients with viral detection had a median PCT level of 0.2 (IQR 0.1–0.7) (Table 8), with 0.4 (IQR 0.2–1.2) for children and 0.1 (IQR 0.1–0.5) for adults. On simulation using the standard PCT cut-off of >0.25 ng/mL to identify bacterial LRTI, having access to PCT test results may

**Table 6. Multivariable logistic regression findings relating to features associated with antibiotic prescription among patients with LRTI and viral detection (n = 438) in Sri Lanka, 2018–2021.**

|  |  | Full model | | Reduced model | |
| --- | --- | --- | --- | --- | --- |
|  |  | OR (95% CI) | P value [a] | OR (95% CI) | P value [a] |
|  | **Children** |  |  |  |  |
| < **0.05** | Neutrophil percentage | 1.04 (1.00–1.07) | 0.06 | 1.04 (1.01–1.07) | **0.01** |
|  | Wheezing | 0.34 (0.09–1.29) | 0.11 | - | - |
|  | **Adults** |  |  |  |  |
| < **0.05** | Headache | 1.40 (0.97–1.91) | **0.04** | 1.39 (0.97–1.91) | **0.04** |
|  | Crepitations/ crackles on exam | 4.90 (2.13–12.64) | <**0.001** | 4.94 (2.15–12.74) | <**0.001** |
|  | Rhonchi/ wheezing on exam | 4.02 (1.73–10.58) | **0.002** | 4.08 (1.76–10.70) | **0.002** |
|  | X-ray opacity/ consolidation | 4.29 (1.27–23.17) | **0.04** | 4.15 (1.25–21.85) | **0.04** |
|  | Cough | 1.15 (0.42–1.81) | 0.65 | - | - |

OR = odds ratio; CI = confidence interval.

[a] P values in bold are significant at <0.05.

**Table 7. Potential reduction in antibiotic prescription when adhering to c-reactive protein (CRP) test results and procalcitonin (PCT) test results among patients with LRTI and viral detection (n = 438) in Sri Lanka, 2018–2021.**

| Adherence level to test results | Potential Reduction in Antibiotic Prescription | | P value [a] |
|---|---|---|---|
| **C-reactive protein (CRP) tests [b]** (Using CRP $\geq$ 20 mg/L for bacterial infection) | **Children** | **Adults** | |
| 60% Adherence | 33.6% | 16.7% | **0.002** |
| 80% Adherence | 44.8% | 22.3% | **<0.001** |
| 100% Adherence | 56.0% | 27.9% | **<0.001** |
| **Procalcitonin (PCT) tests** (Using PCT > 0.25 ng/mL for bacterial infection) | **Children** | **Adults** | **P value** |
| 60% Adherence | 23.4% | 37.0% | **0.04** |
| 80% Adherence | 31.3% | 49.4% | **0.01** |
| 100% Adherence | 39.1% | 61.7% | **0.001** |

CRP = c-reactive protein; PCT = procalcitonin.

[a] P values in bold are significant at <0.05.

[b] We used CRP test results from within 48 hours of hospitalization.

have reduced antibiotic prescription among physicians (Table 7). A level of 60% adherence to PCT-guided antibiotic prescription using PCT >0.25 ng/mL may have potentially reduced inappropriate antibiotic prescription by 23.4% in children and 37.0% in adults; 80% adherence may have potentially reduced inappropriate antibiotic prescription by 31.3% in children and 49.4% in adults; and 100% adherence may have potentially reduced inappropriate antibiotic prescription by 39.1% in children and 61.7% in adults. The potential reduction in antibiotic prescription was significantly greater among adult patients than pediatric patients at 60%, 80%, and 100% adherence levels.

## Discussion

We conducted a prospective cohort study in a large tertiary medical center in Southern Province, Sri Lanka to identify the prevalence and predictors of antibiotic prescription among patients hospitalized with viral LRTI. The goal of this study was to inform future antibiotic stewardship interventions and improve antibiotic prescribing practices. We found a high prevalence of antibiotic prescription among patients with viral detection. Features significantly associated with antibiotic prescription on multivariable analysis included neutrophil percentage among children, and headache, crepitations/ crackles on exam, rhonchi/ wheezing on exam, and x-ray opacity among adults. We showed that access to and adherence to CRP and

**Table 8. Median procalcitonin (PCT) test results among patients with LRTI and viral detection (n = 438) in Sri Lanka, 2018–2021.**

| Procalcitonin (PCT) tests | Children n (%) / median (IQR) | Adults n (%) / median (IQR) | P value [a] |
|---|---|---|---|
| PCT (ng/mL) | 0.4 (0.2–1.2) | 0.1 (0.1–0.5) | 0.9242 |
| PCT $\leq$ 0.1 ng/mL | 14 (17.3) | 114 (42.9) | **<0.001** |
| PCT $\leq$ 0.25 ng/mL | 33 (40.7) | 171 (64.3) | **<0.001** |
| PCT $\leq$ 0.5 ng/mL | 46 (56.8) | 197 (74.1) | **0.002** |

PCT = procalcitonin; IQR = interquartile range.

[a] P values in bold are significant at <0.05.

PCT test results could potentially reduce antibiotic prescriptions by up to 56.0% and 39.1% in children and 27.9% and 61.7% in adults, respectively.

We showed a high percentage of viral detection among a cohort of children and adults hospitalized with LRTI in Sri Lanka, with over one third (36.0%) of enrolled patients having at least one virus detected. A Chinese study found a similar viral detection level of 36.6% among acute LRTI inpatients across 108 hospitals [23]. In contrast, the percentage of viral detection in our study was twice as great as the viral detection level of 15.4% found by a Canadian study among patients hospitalized with respiratory symptoms [24]. Among the viruses detected in the Canadian study were influenza, RSV, parainfluenza, and coronaviruses, all of which are common causes of viral pneumonia [25–27]. While pneumonia was formerly thought to be primarily caused by bacteria, recent studies have increasingly recognized that a high proportion of pneumonia conditions are caused by viruses [4, 28]. No patients were found to test positive for COVID-19 in our study, which could be attributed to the early pandemic practices in Sri Lanka, when patients with COVID-19 were being admitted to specialized isolation hospitals.

Despite finding a high level of viral detection in this study, the results also showed a high proportion of antibiotic prescription (86.8%) among patients with viral detection; therefore, a majority of the antibiotic prescriptions were likely unnecessary. This high prevalence of potential overprescription was found to be nearly equivalent among children and adults (84.4% vs 87.8%), respectively. Similarly, the aforementioned Canadian study found a high prevalence of antibiotic prescription (94.5%) among adult inpatients with viral detection, and thus suggested a positive viral nasopharyngeal test result was not associated with decreased antibiotic prescription [24]. Further, a study in the Netherlands and Israel found that children were significantly less likely to receive an antibiotic prescription than adults (37% vs 83%, p<0.001) among patients at emergency departments or hospitalized with viral RTI [29].

We examined associations between 89 distinct patient features and antibiotic prescription among pediatric and adult patients with viral detection. We found high antibiotic prescription prevalence among children and adults presenting with fever and cough, which are common LRTI symptoms [4]; yet, neither symptom was significantly associated with antibiotic prescription in our study. In contrast, a 2019 study from Malta found acute RTI patients presenting with fever (OR 2.6, 95% CI 2.08–3.26, p<0.001) or productive cough (OR 1.3, 95% CI 1.03–1.61, p = 0.03) were significantly more likely to be prescribed antibiotics than other patients [30]. Headache, also a common LRTI symptom, was found to be significantly associated with antibiotic prescription among adults in the reduced multivariable model. A 2016 Polish study reported an association of headaches while bending forward with antibiotic prescription among pediatric and adult outpatients with RTI (OR 1.93, 95% CI 1.26–2.95, p = 0.003) [31]. However, a 2006 Netherlands study reported headaches were not significantly associated with antibiotic prescription among adult outpatients with LRTIs [32]. Crepitations/ crackles and rhonchi/ wheezing on exam were found to be significantly associated with antibiotic prescription in the reduced adult multivariable model. The previously mentioned 2016 Polish study also reported similar significant associations of rales/ crepitations (OR 26.90, 95% CI 9.00–80.35, p<0.001) and rhonchi/ wheezing (OR 13.03, 95% CI 7.14–23.80, p<0.001) with antibiotic prescription [31]. However, the 2006 Netherlands study found wheezing was not significantly associated with antibiotic prescription among adult outpatients with LRTIs [32]. Neutrophil percentage was the only patient feature that was significantly associated with antibiotic prescription in the children's multivariable model; no associations with neutrophil percentage were found among similar studies. We also found a significant association between x-ray opacity/ consolidation and antibiotic prescription in the reduced model for adults. Similarly, a 2021 Scottish study found abnormal x-ray results were significantly associated with

antibiotic prescription (OR 1.88, 95% CI 1.22–2.90, p = 0.005) among adult outpatients with RTI; however, x-ray abnormality types, such as opacity, were not specified [33]. Although clinical guidelines recommend x-ray testing for LRTI patients [4], there is limited literature that report on the associations of x-ray results and antibiotic prescription.

Despite finding five features that were significantly associated with antibiotic prescription among children and adults, patients without these features also had high rates of antibiotic prescription. For example, patients who presented with cough were significantly more likely to be prescribed antibiotics than those who were not; yet 70.6% of those who did not present with cough were also prescribed antibiotics. Therefore, the patient features significantly associated with antibiotic prescription in this study did not aid in distinguishing whether patients had bacterial or viral LRTI infections. However, our team also conducted PCT tests for research purposes, which could have potentially helped decrease inappropriate antibiotic prescription in this study by 83.3%.

Past studies have suggested biomarker tests such as CRP and PCT may help physicians distinguish bacterial from viral LRTI [10]. When physicians are uncertain about the cause of LRTI, clinical guidelines discourage antibiotic prescription for LRTI patients if their CRP levels are < 20 mg/L or PCT levels are ≤0.25 ng/mL, which show a bacterial infection is unlikely; antibiotic prescription is encouraged if CRP ≥20 mg/L or PCT >0.25 ng/mL, as this points to a high probability of bacterial infection [10]. CRP data were collected as part of clinical testing for 68.5% of the patients with viral detection and were made available to physicians; 38.0% of the patients had low CRP results (0.20 mg/L), and 89.5% of these patients with low CRP results were prescribed antibiotics. We also performed PCT testing for 79.2% of the patients with viral detection; over half of the patients (58.8%) had low PCT results (≤0.25 ng/mL), and 83.3% of these patients with low PCT results were prescribed antibiotics. In contrast, 90.2% of patients with high PCT results (>0.25 ng/mL) received antibiotic prescriptions, as would be recommended by clinical guidelines. The physicians in this study did not have access to PCT test results prior to or during the time of clinical decision making; however, the high level of potential inappropriate antibiotic prescription may have been potentially reduced by more than 80% if the physicians had had access to PCT testing results and followed clinical guidelines. Past studies have found PCT-guided care may lead to physicians to shorten or discontinue antibiotic therapy; it has been associated with a reduction in total antibiotic exposure of 2.9 days among patients with pneumonia [34] and 2.4 days among patients with general LRTI [35], and a reduction in overall antibiotic prescription of 32.4% among patients with pneumonia [36] and between 25% [37] and 47% [38] among patients with general LRTI.

While biomarker testing, such as that of PCT, may be used as tools to guide decision-making on antibiotic therapy for patients with LRTI, physicians are not recommended to use this data as the sole evidence to initiate or end antibiotic therapy [39]. Biomarker testing does not replace clinical judgement or other laboratory tests, and is recommended to supplement other available evidence, such as microbiological data [39]. For this reason, adequate training, such as that offered through antibiotic stewardship, is needed for the optimal use of PCT and other biomarker data [40]. However, it is important to consider that adherence to biomarker-guided antibiotic stewardship training is usually left to the voluntary discretion of physicians [41]; despite having access to CRP results, physicians still overprescribed antibiotics in this study.

Biomarker-guided antibiotic stewardship in LMICs may lead to long-term benefits, such as increased overall cost-savings of hospitals and decreased antimicrobial prescription [40, 42]; however, there are existing barriers to implementing biomarker-guided antibiotic stewardship in LMICs. A 2021 Sri Lankan study reported that despite most physicians stating PCT testing could be helpful, 41% had limited or no prior clinical experience with PCT [43]. In another study, PCT use experts agreed that PCT-guided antibiotic stewardship could help fight acute

RTIs in Asia-Pacific countries; however, they expressed concern for a lack of PCT education among physicians [42]. Thus, increased physician training is required to successfully expand PCT testing in LMICs. Moreover, increased training is required for healthcare authorities and administration to adequately assess the long-term cost-effectiveness of PCT-guided antibiotic stewardship in LMIC hospitals [40].

Strengths of this study include the large sample size and unbiased, consecutive sampling over a long duration, allowing for more precise estimates of the associations between patient features and antibiotic prescription. Further, the study population included pediatric and adult patients, which improved representativeness of the results. Our study had a few limitations. Data regarding ESR and CRP biomarkers were only collected when conducted as part of clinical testing. Biomarkers, such as CRP, have been used as tools to diagnose infections and guide antibiotic prescription [40, 44]; however, past studies have shown PCT levels have greater specificity and sensitivity in detecting bacterial infections than CRP. Due to the limited availability of CRP data in our study, we were unable to determine whether CRP was significantly associated with antibiotic prescription. In this study, we also assumed detection of a respiratory viral pathogen was equivalent to infection, however asymptomatic colonization can be seen with viral organisms. Following standard procedure, we tested nasopharyngeal swabs to diagnose etiology instead of utilizing invasive methods such as bronchoalveolar lavage (BAL); however, past studies have shown that there is substantial concordance in the detection of respiratory viruses between nasopharyngeal swabs and BAL when using multiplex polymerase chain reaction tests [45, 46]. Furthermore, we did not account for the presence of bacterial co-infection. Moreover, this study was conducted at a single center in Southern Province, Sri Lanka. While the study population includes pediatric and adult patients, the generalizability of the results is limited to this center and local area, therefore limiting our ability to expand future interventions to other settings. Further, this study was conducted at a tertiary care hospital and may not reflect care at other lower-level hospitals in Sri Lanka and other LMICs.

## Conclusions

Among a large inpatient cohort with LRTI, viral infection was most common and antibiotic prescription was high. We identified neutrophil percentage, headache, crepitations/ crackles, rhonchi/ wheezing, and x-ray opacity to be associated with antibiotic prescription. CRP and PCT testing, if available, has potential to reduce inappropriate antibiotic prescription by up to 89.5% and 83.3% among inpatients with viral LRTI, respectively. Further research needs to be conducted on the impact of CRP and PCT biomarker testing on antibiotic prescription in low-resource settings to properly inform future antibiotic stewardship interventions.

## Acknowledgments

We would like to acknowledge the great contributions and support from the researchers, physicians, collaborators, study staff, and laboratory personnel of RISE-SL and Ruhuna-Duke Centre for Infectious Diseases. We would also like to acknowledge the support of bioMérieux for providing the procalcitonin kits used in this study.

## Author Contributions

**Conceptualization:** Champica K. Bodinayake, L. Gayani Tillekeratne.

**Data curation:** Nayani Weerasinghe, Sky Vanderburg, Gaya B. Wijayaratne, Buddhika Dilshan, Jack Anderson, Bradly P. Nicholson.

**Formal analysis:** Perla G. Medrano, Lawrence P. Park, Tianchen Sheng.

**Funding acquisition:** Sky Vanderburg, L. Gayani Tillekeratne.

**Investigation:** Perla G. Medrano, Nayani Weerasinghe, Ajith Nagahawatte, Sky Vanderburg, Gaya B. Wijayaratne, Vasantha Devasiri, Tianchen Sheng, Christopher W. Woods, Champica K. Bodinayake, L. Gayani Tillekeratne.

**Methodology:** Perla G. Medrano, Nayani Weerasinghe, Ajith Nagahawatte, Sky Vanderburg, Lawrence P. Park, Gaya B. Wijayaratne, Vasantha Devasiri, Buddhika Dilshan, Tianchen Sheng, Ruvini Kurukulasooriya, Jack Anderson, Bradly P. Nicholson, Christopher W. Woods, Champica K. Bodinayake, L. Gayani Tillekeratne.

**Project administration:** Buddhika Dilshan, Ruvini Kurukulasooriya.

**Resources:** Nayani Weerasinghe, Ajith Nagahawatte, Christopher W. Woods, Champica K. Bodinayake, L. Gayani Tillekeratne.

**Supervision:** Sky Vanderburg, Lawrence P. Park, Bradly P. Nicholson, Christopher W. Woods, Champica K. Bodinayake, L. Gayani Tillekeratne.

**Writing – original draft:** Perla G. Medrano.

**Writing – review & editing:** Perla G. Medrano, Nayani Weerasinghe, Ajith Nagahawatte, Sky Vanderburg, Lawrence P. Park, Gaya B. Wijayaratne, Vasantha Devasiri, Buddhika Dilshan, Tianchen Sheng, Ruvini Kurukulasooriya, Jack Anderson, Bradly P. Nicholson, Christopher W. Woods, Champica K. Bodinayake, L. Gayani Tillekeratne.

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
