## [Decision Letter · Decision Letter 0]

15 Feb 2024

PONE-D-23-38844Prevalence and Predictors of Antibiotic Prescription Among Patients Hospitalized with Viral Lower Respiratory Tract Infections in Southern Province, Sri LankaPLOS ONE

Dear Dr. Medrano, 

Thank you for submitting your manuscript to PLOS ONE. After careful consideration, we feel that it has merit but does not fully meet PLOS ONE’s publication criteria as it currently stands. Therefore, we invite you to submit a revised version of the manuscript that addresses the points raised during the review process. Please submit your revised manuscript by Mar 31 2024 11:59PM. If you will need more time than this to complete your revisions, please reply to this message or contact the journal office at plosone@plos.org. Please include the following items when submitting your revised manuscript:A rebuttal letter that responds to each point raised by the academic editor and reviewer(s). You should upload this letter as a separate file labeled 'Response to Reviewers'.A marked-up copy of your manuscript that highlights changes made to the original version. You should upload this as a separate file labeled 'Revised Manuscript with Track Changes'.An unmarked version of your revised paper without tracked changes. You should upload this as a separate file labeled 'Manuscript'.

We look forward to receiving your revised manuscript.

Kind regards,

Benjamin M. Liu, MBBS, PhD, D(ABMM), MB(ASCP)

Academic Editor

PLOS ONE

Journal Requirements:

3. In the online submission form, you indicated that we are not able to make the data publicly available due to lack of approval to do so by the ethics boards. However, if data are requested, the authors can inquire from the ethics boards at that time.

Additional Editor Comments:

Editor's comments:

1. Introduction: "LRTI is caused by bacterial and viral pathogens; however, viral LRTI is often more commonly detected than bacterial LRTI among both children and adults". The authors missed fungal infection in LRTI and should add fungal infection. The authors should give the readers an overview of the clinical significance, consequences of LRTI or respiratory infections (e.g., CNS infection, sepsis) and diagnostic tools with BAL vs NP. More references should be cited, with the following one as an example (citing suggestion is optional):

Development and Evaluation of a Fully Automated Molecular Assay Targeting the Mitochondrial Small Subunit rRNA Gene for the Detection of Pneumocystis jirovecii in Bronchoalveolar Lavage Fluid Specimens. J Mol Diagn. 2020 Dec;22(12):1482-1493. doi: 10.1016/j.jmoldx.2020.10.003. Epub 2020 Oct 15. Erratum in: J Mol Diagn. 2021 Apr;23(4):506. PMID: 33069878.

2. Introduction: the significance of testing biomarkers, e.g., CRP, PCT and cytokines, should be given. More references should be cited, with the following one as an example (citing suggestion is optional):

Clinical significance of measuring serum cytokine levels as inflammatory biomarkers in adult and pediatric COVID-19 cases: A review. Cytokine. 2021 Jun;142:155478. doi: 10.1016/j.cyto.2021.155478. Epub 2021 Feb 23. PMID: 33667962; PMCID: PMC7901304.

3. Methods: Luminex NxTAG Respiratory Pathogen Panel is not suitable for the diagnosis for LRTI as it tests NP swabs rather than BAL. LRTI or "lower respiratory" used in the manuscript should be changed to "respiratory", in other word, please remove "lower" across the manuscript.

4. Methods: It would be better to explain why Luminex NxTAG Respiratory Pathogen Panel cannot differentiate rhinovirus/enterovirus, to avoid any confusion. More references should be cited, with the following one as an example (citing suggestion is optional):

Universal PCR Primers Are Critical for Direct Sequencing-Based Enterovirus Genotyping. J Clin Microbiol. 2016 Dec 28;55(1):339-340. doi: 10.1128/JCM.01801-16. PMID: 28031445; PMCID: PMC5228251.

Reviewers' comments:

Reviewer's Responses to Questions

**Comments to the Author**

1. Is the manuscript technically sound, and do the data support the conclusions?

Reviewer #1: Yes

Reviewer #2: Yes

2. Has the statistical analysis been performed appropriately and rigorously? 

Reviewer #1: Yes

Reviewer #2: Yes

3. Have the authors made all data underlying the findings in their manuscript fully available?

Reviewer #1: Yes

Reviewer #2: Yes

4. Is the manuscript presented in an intelligible fashion and written in standard English?

Reviewer #1: Yes

Reviewer #2: Yes

5. Review Comments to the Author

Reviewer #1: The article is very interesting and is applicable for the hospital requirements .

I recommend to complete your research and include fungal infection in your research

I recommend to include the results of culture and sensitivity of the antibiotics to confirm the hypothesis.

I recommend to include the PCR results and the primers used

Reviewer #2: Reviewer's Comments to the Author

The study investigated and attempted to identify the prevalence and predictors of antibiotic prescription among children and adults hospitalized with viral Lower Respiratory Tract Infections (LRTI) in a single tertiary health center in Sri Lanka.

The study focus and type were well within the scope of the journal.

The concept of the study is germane, and the expected contribution to knowledge is also good. The methods used were appropriate, while considerable analyses of collated data and relevant statistical analyses were done. The manuscript was also concisely written.

1. Title

The title is appropriate and manages to capture the essence of the study.

2. Abstract:

The Abstract section was concise and well-written. The statistical methods employed were also included in the method segment of the abstract.

3. The introduction:

This section was generally well-written and captures the background of the study appropriately.

4. Materials and Methods:

This section was concisely written to allow for clear understanding and even reproducibility. The study design and sample size were adequate. Inclusion criteria for participants were clearly stated, and the structure of the questionnaire employed was highlighted.

5. Results–

The result section was explicit, and a lot of the data generated was highlighted, and comprehensively expressed in the result section. However, the statement “this could be attributed to the early pandemic practices in Sri Lanka, where patients with COVID-19 were

being admitted to specialized isolation hospitals.......” in lines 168 - 169 should be inserted in the appropriate section in the discussion segment.

Again, on line 178, the demographics of children and adult males were accounted for, but no mention was made of female participants. The demographic demarcation should be properly defined as children and adults, and all adult participants either male or female should be accounted for as adults.

6. Discussion

All inferences expressed under results were comprehensively addressed and discussed in the discussion section. The results were related appropriately to the discussion to capture the findings of the research work. The discussion was done vis-a-vis other related studies to reach a balanced conclusion. As it is, the results aligned completely with the discussion.

6. PLOS authors have the option to publish the peer review history of their article (what does this mean?). If published, this will include your full peer review and any attached files.

Reviewer #1: **Yes: **Professor/ Michael Nazmy Agban - Professor of Microbiology and Immunology faculty of medicine assiut university Egypt

Reviewer #2: **Yes: **ADEYEMI Folasade Muibat

---

## [Author Response · Author response to Decision Letter 0]

30 Apr 2024

Thank you for considering our revised manuscript entitled “Prevalence and Predictors of Antibiotic Prescription Among Patients Hospitalized with Viral Lower Respiratory Tract Infections in Southern Province, Sri Lanka” for publication in PLOS ONE. We appreciate the detailed comments of the reviewers and editors, and have made every attempt to address them.

Journal Requirements:

1. When submitting your revision, we need you to address these additional requirements. Please ensure that your manuscript meets PLOS ONE's style requirements, including those for file naming. The PLOS ONE style templates can be found at 

We revised the manuscript to meet PLOS ONE’s style requirements.

2. Update the submission using a PLOS LaTeX template. The template and more information on our requirements for LaTeX submissions can be found at http://journals.plos.org/plosone/s/latex. 

We have started the process of updating the manuscript using a PLOS LaTeX template. We will upload a LaTeX source file when it is requested.

3. In the online submission form, you indicated that we are not able to make the data publicly available due to lack of approval to do so by the ethics boards. However, if data are requested, the authors can inquire from the ethics boards at that time. All PLOS journals now require all data underlying the findings described in their manuscript to be freely available to other researchers, either a. In a public repository, b. Within the manuscript itself, or c. Uploaded as supplementary information. This policy applies to all data except where public deposition would breach compliance with the protocol approved by your research ethics board. If your data cannot be made publicly available for ethical or legal reasons (e.g., public availability would compromise patient privacy), please explain your reasons on resubmission and your exemption request will be escalated for approval.

We have revised our explanation in the resubmission to include: “We are not able to make the data publicly available as public deposition would breach compliance with the protocol approved by the research ethics board.”

Additional Editor Comments:

*Note: Due to a known limitation with Microsoft Word's track changes 'all markup' view, some line numbers in the copy with tracked changes may not consistently align with those in the clean, unmarked manuscript copy. This issue can cause several line numbers to be skipped or mismatched in the 'all markup' view. To ensure clarity and ease of reference, all line number mentions in our responses correspond to the line numbers in the clean, unmarked manuscript. Should you wish to cross-reference these revisions in the tracked changes copy, we recommend viewing the document in 'simple markup' view, where the correct line numbers are accurately displayed. We appreciate your understanding and please do not hesitate to contact us should you need further clarification on any of the changes.

We would like to address the comments provided by Academic Editor Benjamin M. Liu:

Comment #1. Introduction: "LRTI is caused by bacterial and viral pathogens; however, viral LRTI is often more commonly detected than bacterial LRTI among both children and adults". The authors missed fungal infection in LRTI and should add fungal infection. The authors should give the readers an overview of the clinical significance, consequences of LRTI or respiratory infections (e.g., CNS infection, sepsis) and diagnostic tools with BAL vs NP. More references should be cited, with the following one as an example (citing suggestion is optional): Development and Evaluation of a Fully Automated Molecular Assay Targeting the Mitochondrial Small Subunit rRNA Gene for the Detection of Pneumocystis jirovecii in Bronchoalveolar Lavage Fluid Specimens. J Mol Diagn. 2020 Dec;22(12):1482-1493. doi: 10.1016/j.jmoldx.2020.10.003. Epub 2020 Oct 15. Erratum in: J Mol Diagn. 2021 Apr;23(4):506. PMID: 33069878.

We appreciate this thoughtful and important comment from the editor. We have revised Lines 60-61 to state that LRTI is caused primarily by bacterial and viral pathogens, and less commonly by fungal organisms. We also added Line(s) 66-69 and 77-86 to provide an overview of consequences of LRTI if untreated and how LRTI is diagnosed. We used the following reference(s):

4. Dasaraju PV, Liu C. Infections of the Respiratory System. Nih.gov. University of Texas Medical Branch at Galveston; 2014. Available from: https://www.ncbi.nlm.nih.gov/books/NBK8142/

5. Woodhead M, Blasi F, Ewig S, Garau J, Huchon G, Ieven M, et al. Guidelines for the management of adult lower respiratory tract infections - Full version. Clinical Microbiology and Infection [Internet]. 2011 Nov 26;17:E1–59. Available from: https://www.ncbi.nlm.nih.gov/pmc/articles/PMC7128977/

10. Noviello S, Huang D. The Basics and the Advancements in Diagnosis of Bacterial Lower Respiratory Tract Infections. Diagnostics [Internet]. 2019 Apr 3;9(2):37. Available from: https://www.ncbi.nlm.nih.gov/pmc/articles/PMC6627325/

13. Wijnands GJA. Diagnosis and interventions in lower respiratory tract infections. The American Journal of Medicine [Internet]. 1992 Apr 6;92(4):S91–7. Available from: https://www.amjmed.com/article/0002-9343(92)90317-5/abstract 

14. Carroll KC. Laboratory Diagnosis of Lower Respiratory Tract Infections: Controversy and Conundrums. Journal of Clinical Microbiology [Internet]. 2002 Sep 1;40(9):3115–20. Available from: https://www.ncbi.nlm.nih.gov/pmc/articles/PMC130746/

Comment #2. Introduction: the significance of testing biomarkers, e.g., CRP, PCT and cytokines, should be given. More references should be cited, with the following one as an example (citing suggestion is optional): Clinical significance of measuring serum cytokine levels as inflammatory biomarkers in adult and pediatric COVID-19 cases: A review. Cytokine. 2021 Jun;142:155478. doi: 10.1016/j.cyto.2021.155478. Epub 2021 Feb 23. PMID: 33667962; PMCID: PMC7901304.

We appreciate this thoughtful and important comment from the editor. We added Line(s) 84-86 to state that biomarker testing may help shed light on LRTI etiology.

Comment #3. Methods: Luminex NxTAG Respiratory Pathogen Panel is not suitable for the diagnosis for LRTI as it tests NP swabs rather than BAL. LRTI or "lower respiratory" used in the manuscript should be changed to "respiratory", in other word, please remove "lower" across the manuscript.

We appreciate this thoughtful and important comment from the editor. We felt that the Luminex NxTAG Respiratory Pathogen Panel is suitable for the diagnosis for LRTI as it is standard to use NP swabs to diagnose etiology and obtaining BALs is invasive. We have added Lines 143-146 to justify why we used the Luminex NxTAG Respiratory Pathogen Panel NP swabs and listed this is as a limitation in the discussion section in Lines 455-458. Furthermore, we felt that using LRTI or "lower respiratory" in the manuscript is appropriate as we used a clinical case definition for LRTI that has been used by other seminal studies, such as those cited below. We have therefore chosen to keep "lower respiratory" across the manuscript.

6. O’Brien KL, Baggett HC, Brooks WA, Feikin DR, Hammitt LL, Higdon MM, et al. Causes of severe pneumonia requiring hospital admission in children without HIV infection from Africa and Asia: the PERCH multi-country case-control study. The Lancet. 2019 Aug;394(10200):757–79. Available from: https://www.ncbi.nlm.nih.gov/pmc/articles/PMC6727070/#!po=15.0000

7. Jain S, Self WH, Wunderink RG, Fakhran S, Balk R, Bramley AM, et al. Community-Acquired Pneumonia Requiring Hospitalization among U.S. Adults. New England Journal of Medicine. 2015 Jul 30;373(5):415–27. Available from: https://www.ncbi.nlm.nih.gov/pmc/articles/PMC4728150/

8. Jain S, Williams DJ, Arnold SR, Ampofo K, Bramley AM, Reed C, et al. Community-Acquired Pneumonia Requiring Hospitalization among U.S. Children. New England Journal of Medicine. 2015 Feb 26;372(9):835–45. Available from: https://www.nejm.org/doi/full/10.1056/NEJMoa1405870

Comment #4. Methods: It would be better to explain why Luminex NxTAG Respiratory Pathogen Panel cannot differentiate rhinovirus/enterovirus, to avoid any confusion. More references should be cited, with the following one as an example (citing suggestion is optional): Universal PCR Primers Are Critical for Direct Sequencing-Based Enterovirus Genotyping. J Clin Microbiol. 2016 Dec 28;55(1):339-340. doi: 10.1128/JCM.01801-16. PMID: 28031445; PMCID: PMC5228251.

We appreciate this thoughtful and important comment from the editor. We have added Lines 141-143 to explain why Luminex NxTAG Respiratory Pathogen Panel cannot differentiate rhinovirus/enterovirus and cited the additional reference below.

19. NxTAG ® Respiratory Pathogen Panel + SARS-CoV-2 Package Insert [Internet]. Luminex Corporation. U.S. Food and Drug Administration; 2022 Nov. Available from: https://www.fda.gov/media/146495/download

We would like to address the comments provided by Reviewer #1 Michael Nazmy Agba:

Comment #1: The article is very interesting and is applicable for the hospital requirements. I recommend to complete your research and include fungal infection in your research. 

We thank the reviewer for the suggestions to improve our manuscript. We have revised Lines 60-61 in the introduction to say that LRTI is caused primarily by bacterial and viral pathogens, and less commonly by fungal organisms.

Comment #2: I recommend to include the results of culture and sensitivity of the antibiotics to confirm the hypothesis. 

We also added Line(s) 66-69 and 77-86 to provide an overview of consequences of LRTI if untreated and how LRTI is diagnosed.

Comment #3: I recommend to include the PCR results and the primers used

Regarding PCR results and primers, we have cited the test (Luminex NxTAG Respiratory Pathogen Panel; Luminex Corporation, Austin, TX, USA) which uses proprietary technology.

18. NxTAG® Respiratory Pathogen Panel Test | Diasorin [Internet]. int.diasorin.com. DiaSorin; Available from: https://int.diasorin.com/en/molecular-diagnostics/kits-reagents/nxtag-respiratory-pathogen-panel

19. NxTAG ® Respiratory Pathogen Panel + SARS-CoV-2 Package Insert [Internet]. Luminex Corporation. U.S. Food and Drug Administration; 2022 Nov. Available from: https://www.fda.gov/media/146495/download

We would like to address the comments provided by Reviewer #2 Adeyemi Folasade Muibat:

Comment #1: The result section was explicit, and a lot of the data generated was highlighted, and comprehensively expressed in the result section. However, the statement “this could be attributed to the early pandemic practices in Sri Lanka, where patients with COVID-19 were being admitted to specialized isolation hospitals.......” in lines 168 - 169 should be inserted in the appropriate section in the discussion segment. 

We thank the reviewer for the suggestions to improve our manuscript. We have moved the statement into the discussion section in Lines 360-363. 

Comment #2: Again, on line 178, the demographics of children and adult males were accounted for, but no mention was made of female participants. The demographic demarcation should be properly defined as children and adults, and all adult participants either male or female should be accounted for as adults.

We thank the reviewer for the suggestions to improve our manuscript. We have revised Line 182 in the results to explicitly mention the demographics of male and female participants.

---

## [Decision Letter · Decision Letter 1]

17 May 2024

Prevalence and Predictors of Antibiotic Prescription Among Patients Hospitalized with Viral Lower Respiratory Tract Infections in Southern Province, Sri Lanka

PONE-D-23-38844R1

Dear Dr. Medrano,

We’re pleased to inform you that your manuscript has been judged scientifically suitable for publication and will be formally accepted for publication once it meets all outstanding technical requirements.

Kind regards,

Benjamin M. Liu, MBBS, PhD, D(ABMM), MB(ASCP)

Academic Editor

PLOS ONE

Additional Editor Comments (optional):

Reviewers' comments:

Reviewer's Responses to Questions

**Comments to the Author**

1. If the authors have adequately addressed your comments raised in a previous round of review and you feel that this manuscript is now acceptable for publication, you may indicate that here to bypass the “Comments to the Author” section, enter your conflict of interest statement in the “Confidential to Editor” section, and submit your "Accept" recommendation.

Reviewer #1: All comments have been addressed

2. Is the manuscript technically sound, and do the data support the conclusions?

Reviewer #1: Yes

3. Has the statistical analysis been performed appropriately and rigorously? 

Reviewer #1: Yes

4. Have the authors made all data underlying the findings in their manuscript fully available?

Reviewer #1: Yes

5. Is the manuscript presented in an intelligible fashion and written in standard English?

Reviewer #1: Yes

6. Review Comments to the Author

Reviewer #1: I'm so glad that you could address all the comments and fulfilled all the requirements asked before.

7. PLOS authors have the option to publish the peer review history of their article (what does this mean?). If published, this will include your full peer review and any attached files.

Reviewer #1: **Yes: **Dr Michael Nazmy Agban professor of Microbiology and Immunology faculty of medicine assiut university Egypt

---

## [Editor Report · Acceptance letter]

30 May 2024

PONE-D-23-38844R1 

PLOS ONE

Dear Dr. Medrano, 

I'm pleased to inform you that your manuscript has been deemed suitable for publication in PLOS ONE. Congratulations! Your manuscript is now being handed over to our production team.

Kind regards, 

on behalf of

Dr. Benjamin M. Liu 

Academic Editor

PLOS ONE